⊝ | **Editor's Pick** | Clinical Microbiology | Research Article

# Uncharacterized and lineage-specific accessory genes within the *Proteus mirabilis* pan-genome landscape

Robert F. Potter,[1,2] Kailun Zhang,[2,3] Ben Reimler,[2] Jamie Marino,[4] Carol E. Muenks,[2] Kelly Alvarado,[2] Meghan A. Wallace,[2] Lars F. Westblade,[4] Erin McElvania,[5] Melanie L. Yarbrough,[2] David A. Hunstad,[1,6] Gautam Dantas,[1,2,3,6,7] Carey-Ann D. Burnham[1,2,6,8]

**ABSTRACT** *Proteus mirabilis* is a Gram-negative bacterium recognized for its unique swarming motility and urease activity. A previous proteomic report on four strains hypothesized that, unlike other Gram-negative bacteria, *P. mirabilis* may not exhibit significant intraspecies variation in gene content. However, there has not been a comprehensive analysis of large numbers of *P. mirabilis* genomes from various sources to support or refute this hypothesis. We performed comparative genomic analysis on 2,060 *Proteus* genomes. We sequenced the genomes of 893 isolates recovered from clinical specimens from three large US academic medical centers, combined with 1,006 genomes from NCBI Assembly and 161 genomes assembled from Illumina reads in the public domain. We used average nucleotide identity (ANI) to delineate species and subspecies, core genome phylogenetic analysis to identify clusters of highly related *P. mirabilis* genomes, and pan-genome annotation to identify genes of interest not present in the model *P. mirabilis* strain HI4320. Within our cohort, *Proteus* is composed of 10 named species and 5 uncharacterized genomospecies. *P. mirabilis* can be subdivided into three subspecies; subspecies 1 represented 96.7% (1,822/1,883) of all genomes. The *P. mirabilis* pan-genome includes 15,399 genes outside of HI4320, and 34.3% (5,282/15,399) of these genes have no putative assigned function. Subspecies 1 is composed of several highly related clonal groups. Prophages and gene clusters encoding putatively extracellular-facing proteins are associated with clonal groups. Uncharacterized genes not present in the model strain *P. mirabilis* HI4320 but with homology to known virulence-associated operons can be identified within the pan-genome.

**IMPORTANCE** Gram-negative bacteria use a variety of extracellular facing factors to interact with eukaryotic hosts. Due to intraspecies genetic variability, these factors may not be present in the model strain for a given organism, potentially providing incomplete understanding of host-microbial interactions. In contrast to previous reports on *P. mirabilis*, but similar to other Gram-negative bacteria, *P. mirabilis* has a mosaic genome with a linkage between phylogenetic position and accessory genome content. *P. mirabilis* encodes a variety of genes that may impact host-microbe dynamics beyond what is represented in the model strain HI4320. The diverse, whole-genome characterized strain bank from this work can be used in conjunction with reverse genetic and infection models to better understand the impact of accessory genome content on bacterial physiology and pathogenesis of infection.

**KEYWORDS** *Proteus mirabilis*, microbial genomics, population structure

Many bacterial taxa can be classified as opportunistic pathogens, organisms that exhibit context-dependent degrees of commensalism or pathogenicity. The mechanisms that shift an organism from a commensal to a pathogenic lifestyle are likely multifactorial and rely on synergy of microbial and host features; however, such

Address correspondence to Gautam Dantas, dantas@wustl.edu, or Carey-Ann D. Burnham, cburnham@wustl.edu.

Robert F. Potter and Kailun Zhang contributed equally to this article. Author order was determined alphabetically by last name.

The authors declare no conflict of interest.

See the funding table on p. 14.

complexity is overlooked in conventional host-pathogen interaction models that use single strains of bacteria (1). One issue with model strains is that they may lack genetic factors present in currently circulating pathogens that may contribute to phenotypic heterogeneity. Gram-negative bacteria use gene products that span the outer membrane to differentially adhere to and invade eukaryotic host cells, including type one secretion systems, type V secretion systems, and chaperone-usher pili (2–4). These genes can be variably present across strains within bacterial species, as observed for *Klebsiella variicola*, *Klebsiella pneumoniae*, and *Escherichia coli* (5–7). Acquisition of specific genetic factors may result in clonal propagation and spread of successful lineages (8, 9). Genomic characterization of the clonally epidemic *K. pneumoniae* ST15 lineage identified the Kpi chaperone-usher pilus operon, which was found to confer gastrointestinal adherence and was absent from the well-characterized model strain, *K. pneumoniae* ATCC10031 (8). Similarly, whole genome sequencing revealed that one explanation for the global expansion of *Salmonella enterica* serovar Typhimurium clone ST34 is acquisition of the *sopE* type III secretion system effector, enabling increased microbial uptake (9). Accordingly, characterization of bacterial genomic population structure and pan-genome content in Gram-negative opportunistic pathogens has increased our understanding of how these bacteria interact with human hosts beyond what model strains may show.

*P. mirabilis* is an Enterobacterales historically noted for its swarming motility and urease activity (10). Substantial work characterizing the model strain *P. mirabilis* HI4320 has identified several factors at the host-microbe interface that contribute to uropathogenesis, including mannose-resistant *Proteus*-like (MR/P) fimbriae, *Proteus mirabilis* fimbriae (PMF), *Proteus* toxic agglutinin (*pta*), trimeric autoagglutin autotransporter of *Proteus* (*taaP*), and adhesion and invasion autotransporter (*aipA*) (11–13). Despite being a common cause of urinary tract infections (UTIs) and being associated with wound and soft tissue infections, this organism has not been the subject of a large, focused comparative genomic investigation (14, 15). A small study raised the hypothesis that *P. mirabilis* differs from other Gram-negative bacteria in not having a mosaic pan-genome (10). However, this conclusion was drawn from an analysis of only three newly sequenced strains being compared against the *P. mirabilis* HI4320 reference genome (10). Evidence that inclusion of more strains might identify further genetic diversity arose during an investigation of *P. mirabilis*-enhanced Crohn's disease (16). Hierarchical clustering of average nucleotide identity (ANI) analysis of 33 sequenced *P. mirabilis* genomes from that investigation and 24 genomes from NCBI databases found that 95% (54/57) of genomes fell into two groups, 39% (21/54) in one and 61% (33/54) in the other, with their clinical strains interspersed with NCBI genomes (16). Therefore, a gap in knowledge exists regarding the extent of inter-strain diversity of *P. mirabilis,* specifically on the presence of highly related lineages and variability of genetic factors that may interact with eukaryotic host cells.

To address this gap in knowledge, we performed a multicenter analysis of 2,060 *Proteus* genomes. We provide the first description of *P. mirabilis* subspecies, clusters of highly related lineages with similar accessory gene content, and the presence of many uncharacterized genes that could alter *P. mirabilis*-host dynamics. This work highlights the genetic diversity of *P. mirabilis* and shows that, in contrast to prior reports, the intraspecies gene diversity of *P. mirabilis* is similar to other Enterobacterales members. These findings also lay a foundation for future investigation on functional consequences of *P. mirabilis* inter-strain genetic variability at the host-microbe interface.

## MATERIALS AND METHODS

### Isolate cohort

Clinical isolates of *P. mirabilis* were obtained from patient samples processed according to standard clinical procedures for the specimen type at Washington University School of Medicine (St. Louis, Missouri) for microbiologic culture (17). From September 2020

to November 2021 we collected *Proteus* organisms (*n* = 427) that were identified and reported out to the species level as part of the culture results during routine clinical care (i.e., always reported from sterile body sites, such as blood cultures, when pure or predominant in mixed cultures from non-sterile body sites, or above sample-specific threshold for quantitative culture types, such as clean catch urine specimens) in the electronic medical record (EMR) per routine clinical procedure. Non-sterile urine samples were plated to tryptic soy agar with 5% sheep blood (Hardy Diagnostics, Santa Maria, California, USA) and MacConkey agar (Hardy Diagnostics, Santa Maria, California, USA). Sterile urine samples, tissue, wound, respiratory, and positive blood culture bottles were plated to tryptic soy agar with 5% sheep blood, chocolate agar (Hardy Diagnostics, Santa Maria, California, USA) and MacConkey agar. All samples were incubated at 35°C in ambient air for aerobic culture. Additionally, we collected organisms (*n* = 213) in a convenience sampling not identified and reported in the EMR per routine clinical procedure (i.e., isolates that were grouped as part of "mixed microorganisms" in the clinical culture report or were below clinical reporting thresholds) but were suspected to be *Proteus* spp. due to the characteristic odor of the microbe or swarming motility on culture medium. The recovered organisms were identified by MALDI-ToF MS (Bruker Biotyper). Our cohort also included *P. mirabilis* isolates (*n* = 97) that were recovered and reported from sterile body sites from 2017 to 2019 and frozen at −80°C. Finally, our cohort included isolates from human stool (*n* = 3) and from skin swabs of healthy individuals in Pakistan (*n* = 13) collected as part of separate investigations. All organisms in the total cohort were subcultured onto tryptic soy agar with 5% sheep blood prior to archiving to exclude mixed populations. Additional *P. mirabilis* isolates that grew from blood, urine, or tissue/wound samples were obtained from NorthShore University Health System (*n* = 75) and Weill Cornell Medicine (*n* = 61). Organisms were assigned a study number, frozen in tryptic soy broth with 10% glycerol, and stored at −80°C.

## Patient and laboratory metadata

Chi-squared test (*P* < 0.05 for the threshold of significance) was used to determine if there was a statistical association between anatomic source and population structure.

## Draft whole genome sequencing

The aforementioned freezer stocks were subcultured onto tryptic soy agar with 5% sheep blood using inoculating loops. Cultures were incubated for 16–20 h at 35°C in room air; ~10 colonies of non-swarming *Proteus* spp. or a sweep of the fourth quadrant for swarming *Proteus* spp. was suspended in sterile, molecular grade water (Thermo Fisher Scientific, Waltham, Massachusetts, USA). Total genomic DNA was extracted using the Bacteremia Kit (Qiagen, Germantown, Maryland, USA) according to the manufacturer's instructions; 0.5ng of each DNA sample was used to create Illumina sequencing libraries with a modification of the Nextera XT protocol (18). Examples of all computational commands for this study are included (Document S1). Samples were pooled and sequenced on an Illumina NovaSeq platform by the Genome Technology Access Center at McDonnell Genome Institute (https://gtac.wustl.edu/). Raw reads were demultiplexed by barcode and had adapters removed using trimmomatic v.38 (19). Processed reads were assembled into draft genomes with unicycler v1.0 (20). Assembly quality was assessed with QUAST v4.5 (21). Genomes were included in this study if "# contigs (>= 0 bp)" metric from QUAST was below 500 contigs (Table S2). All assemblies that passed quality filtering had genes annotated with prokka v1.14 (22).

## Analysis of publicly available genomes

We acquired publicly available *Proteus* genomes from GenBank in June 2022 (*n* = 1,006). Fasta files for "*Proteus*" genomes were obtained from NCBI Assembly using the GenBank nomenclature in June 2022. Reads for "*Proteus*" that were "paired" library layout, from "DNA" source, "Genome" strategy, and "fastq" File Type were downloaded from NCBI SRA

in June 2022. Downloaded SRA reads ($n = 107$) were processed using trimmomatic and unicycler exactly as described above. Assembled SRA scaffolds and fasta files directly from NCBI Assembly had their quality assessed with QUAST and genes annotated with prokka as described above. Information on genomes used is available in Table S1.

## ANI analysis

All assembly files ($n = 2,060$) that passed quality filtering were used as input for an all-by-all comparison with FastANI (23). The resulting pairwise comparison file ANI values were filtered to remove pairwise comparisons that yielded an ANI below the accepted species cutoff of 95%. The resulting source-target-edge file was input into Cytoscape for visualization (23). Node shapes were altered to correspond with the genome source. Species assessment for the Cytoscape reciprocal groups used the type assembly for all valid *Proteus* spp. The subset consisting of the ANI-confirmed *P. mirabilis* ($n = 1,883$) was visualized as a heatmap in RStudio with a gradient centered at 98% (24). This cutoff was previously used for subspecies delineation of *Salmonella, Mycobacterium abscessus, and Leuconostoc lactis* (25–27). Subspecies classification was defined by Cytoscape reciprocal groups composed of genomes with ANI $\geq$ 98% to other genomes within the group but 98% < ANI $\geq$ 95% between groups.

## Core genome alignment

All ANI-confirmed *P. mirabilis* genomes ($n = 1,883$) had prokka-identified genes clustered using panaroo v1.0 using a core threshold value of 99% under moderate mode (28) . The core genome alignment file was processed using SNP-sites to keep only polymorphic positions (29). The filtered core genome alignment was converted into a newick tree using FastTree v2.1.9 with the gamma flag activated (30). The resulting newick file was visualized as an unrooted phylogenetic tree using the iTOL website (31).

We next used RAxML to identify duplicate isolates from the core genome alignment of all 1,883 subspecies 1 genomes. Duplicates could arise from the use of publicly available data (i.e., authors upload genome and Illumina reads, or authors upload multiple genomes for the same isolate) or our own analysis (if a patient presents with *P. mirabilis* infection at different time points) and may bias our results by artificially inflating the size of clonal clusters. The subset of *P. mirabilis* subspecies 1 genomes ($n = 1,748$) after removal of duplicates was processed using panaroo, SNP-sites, FastTree, and iTOL in the same manner. iTOL was used to overlay metadata with the color strip function and gene presence/absence data as a binary function. We used SNP counting method snp-dists (v0.8.2) (https://github.com/tseemann/snp-dists) to count every pairwise SNP site. The output was converted into source-target-edge (genome A-genome B-SNP Count) file and filtered to remove edges $\leq$ 4,013 SNPs. We wanted to enact an SNP cutoff value stringent enough to define clusters consisting only of highly related genomes, excluding possible singleton genomes, but larger than traditional cutoffs used for defining local outbreaks. We chose 4,013 SNPs as it represents 0.1% of the median genome length of *P. mirabilis* genomes on NCBI Genomes (Accessed July 2022). We used FastBAPS as an additional method for clustering genomes and compared concordance between SNP clusters and BAPS grouping (32). Results depicting each of the two methods for *P. mirabilis* subspecies 1 genome are included (Table S3).

## Pan-genome analysis

The pan_genome_reference fasta file from panaroo analysis of *P. mirabilis* subspecies 1 genomes was uploaded to EggNOG-mapper and annotated with the EggNog five database (33). The binary accessory gene-presence absence matrix from panaroo had singleton (found in one genome) and core genes (found in $\geq$1,731 genomes) removed. The resulting matrix was visualized using Rtsne (https://cran.r-project.org/web/packages/Rtsne/) as a T-Distributed Stochastic Neighbor Embedding (t-SNE) plot. The parameters used two dimensions, 40 perplexity, max iteration of 5,000, and had

check_duplicates turned off. Samples were colored if they came from one of the top 10 clusters as determined by SNP counting. Scoary was used to identify genes significantly enriched within the three largest clusters by using the corresponding annotated-gene presence-absence matrix from panaroo and a traits file where genome presence in any of the three largest clusters was denoted with a one and absence with a zero (34). The results were filtered to include only genes with sensitivity and specificity ≥80%. Genes visualized that were found spatially localized adjunct to one another are described (Table S4). We used a modification of roary_plots (https://github.com/sanger-patho-gens/Roary/blob/master/contrib/roary_plots/roary_plots.ipynb) to visualize the newick tree for subspecies 1 genomes with the presence/absence matrix from panaroo. Selected genes described in Fig. 6 were chosen for display from annotation by panaroo and putative localization as extracellular facing factors. NCBI annotations for the specific genes of interest described in Fig. 6 are included (Table S5).

## RESULTS

### *Proteus* cohort is composed of 10 named species and 5 genomospecies

We initially performed ANI analysis on the 2,060 genomes and clustered pairwise ANI values ≥95% into nodes using Cytoscape (Fig. S1). We found that the 2,060 genomes fell into 15 nodes, with no ANI values ≥95% existing between nodes, indicating robust species delineation within the *Proteus* genus. Analysis of type genomes within the respective nodes revealed that *P. mirabilis* was the largest node, representing 91% ($n$ = 1,883) of the cohort, followed by *Proteus terrae* at 2.5% ($n$ = 51), *Proteus penneri* at 1.7% ($n$ = 36), *Proteus vulgaris* 1.5% ($n$ = 30), and *Proteus columbae* at 1.3% ($n$ = 27). One-third (5/15) of the nodes did not have a type strain within them, indicating they represent novel *Proteus* genomospecies. All novel genomospecies came from the publicly deposited genomes. A midpoint-rooted phylogenetic tree of the 1,101 *Proteus* core genes from type genomes for named species and representative genomes for the novel genomospecies showed that *P. mirabilis* is most closely related to the singleton *Proteus myxofaciens*, both forming a monophyletic group with *Proteus hauseri* (Fig. 1).

### *P. mirabilis* comprises one major subspecies and two minor subspecies

We next visualized the same ANI analysis to examine the 1,883 *P. mirabilis* genomes as a heatmap color gradient centered at 98% ANI (Fig. 2A), a cutoff used previously for subspecies delineation in *Salmonella* (27). Hierarchical clustering of the pairwise ANI values revealed three groups share ≥98% ANI within each group but <98% ANI between groups, indicating that *P. mirabilis* is composed of three subspecies (Fig. 2A). The largest of these, which we term subspecies 1, contained 96.7% ($n$ = 1,822) of the genomes, 2.54% ($n$ = 48) were in subspecies 2, and 0.7% ($n$ = 13) were in subspecies 3. As an orthogonal method to confirm the distinct relationship among the putative *P. mirabilis* subspecies, we constructed an approximate maximum likelihood phylogenetic tree from the alignment of the 2,089 core genes found in >99% of the *P. mirabilis* genomes and found that all subspecies 2 and 3 genomes from the ANI analysis formed monophyletic clades within each subspecies (Fig. 2B). The short branch distance (0.021 for subspecies 2 and 0.088 subspecies 3) to their connecting node and position indicated that subspecies 2 and 3 are more closely related to each other than to subspecies 1. Given that these represented only 3.2% (61/1,883) of the *P. mirabilis* cohort, the remainder of our analysis focused on subspecies 1 genomes.

### A wealth of undescribed genes exist in the *P. mirabilis* pan-genome

We performed panaroo pan-genome analysis on the reduced cohort of 1,748 *P. mirabilis* subspecies 1 genomes to gain insight into gene content and relatedness. This analysis resulted in a total pan-genome size of 19,069 genes. Of these, 2,082 (10.9%) were considered core genes (>99% prevalence), 2,059 (10.8%) were considered shell genes (15–99% prevalence), and 14,928 (78.3%) were considered cloud genes (<15%

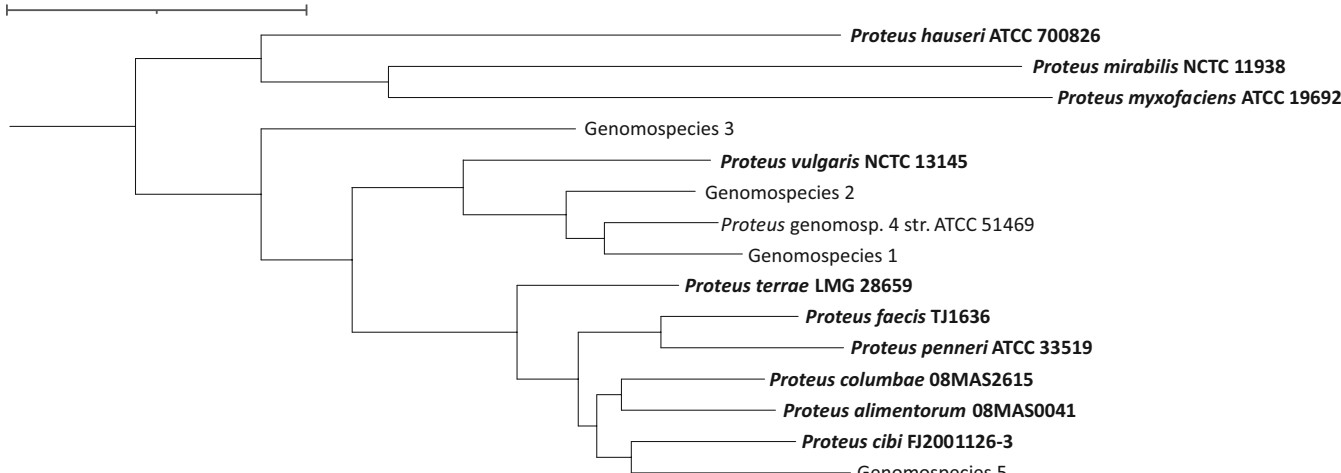

**.1 nucleotide substitution per site**

**FIG 1** *P. mirabilis* is most closely related to *P. myxofaciens* and *P. hauseri*. Core genome phylogenetic tree of type strain genomes for named taxonomic species and representative genomes for uncharacterized genomospecies.

prevalence) (Fig. S2) (35). The pan-genome reference file containing single representatives of all 19,069 genes was uploaded to the EggNOG-mapper annotation website, yielding an annotation for 13,633 genes (71.4%). Clusters of orthologous groups (COGs; *n* = 12,305) were assigned to a smaller subset of 11,495 (60.2%) genes in the pan-genome (Fig. 3A).

Since the early 1990s, most of the knowledge of *P. mirabilis* biology and pathogenesis, particularly in the urinary tract, has come from studies using *P. mirabilis* strain HI4320 (11, 36). We found that 3,670 ORFs were detected by prokka in the *P. mirabilis* HI4320 complete genome (Fig. 3A). EggNOG-mapper assigned an annotation to 3,516 (95.8%) of these, while assigning an annotation to only 10,117 (65.7%) of 15,399 genes not present in *P. mirabilis* HI4320 (*P* < 0.0001; Fig. 3A). We then analyzed the conservation of each gene compared with the *P. mirabilis* HI4320 chromosome to identify regions that were present in the accessory genome (Fig. 3B). We found five regions previously posited to be part of the accessory genome due to GC-skewing during the initial whole-genome sequencing of *P. mirabilis* HI4320 did have variable presence within the *P. mirabilis* pan-genome (37). The least conserved of these was a 70-gene conjugative transposon present in 17% of all *P. mirabilis* strains. We also identified a 24-gene prophage (PMI1906-1930) that was not initially described in the first complete genome *of P. mirabilis* HI4320 (37). Also included in the core genome are the characterized urovirulence factors *pta*, *taap*, and *aipA*, and the chaperone-usher pili PMF and MR/P. We next applied pfam annotations to the 15,399 genes present in the *P. mirabilis* pan-genome outside of HI4320. Interestingly, the three most highly represented pfam categories among non-HI4320 genes all had motifs involved in DNA interactions (HNH_3, phage_integrase, and rve); moreover, five additional annotations involved in DNA interactions (HTH_Tnp_1, ResIII, Helicase_C, HTH_21, and Arm-DNA-bind_3) were represented within the top 20 pfam categories. Meanwhile, non-HI4320 genes involved with putative extracellular-facing interactions comprised seven of the top 20 pfam categories, specifically including glycosylation (Glycos_transf_1, Glycos_transf_2, and Glycos_transf_4), transport (ABC_tran, MFS_1), and adhesion/virulence (fimbrial, PAAR_motif) (Fig. 3C).

## *P. mirabilis* accessory genome composition is associated with phylogenetic position

We created an approximate maximum-likelihood SNP core genome phylogenetic tree of the 1,748 *P. mirabilis* subspecies 1 genomes to visualize the population structure of our

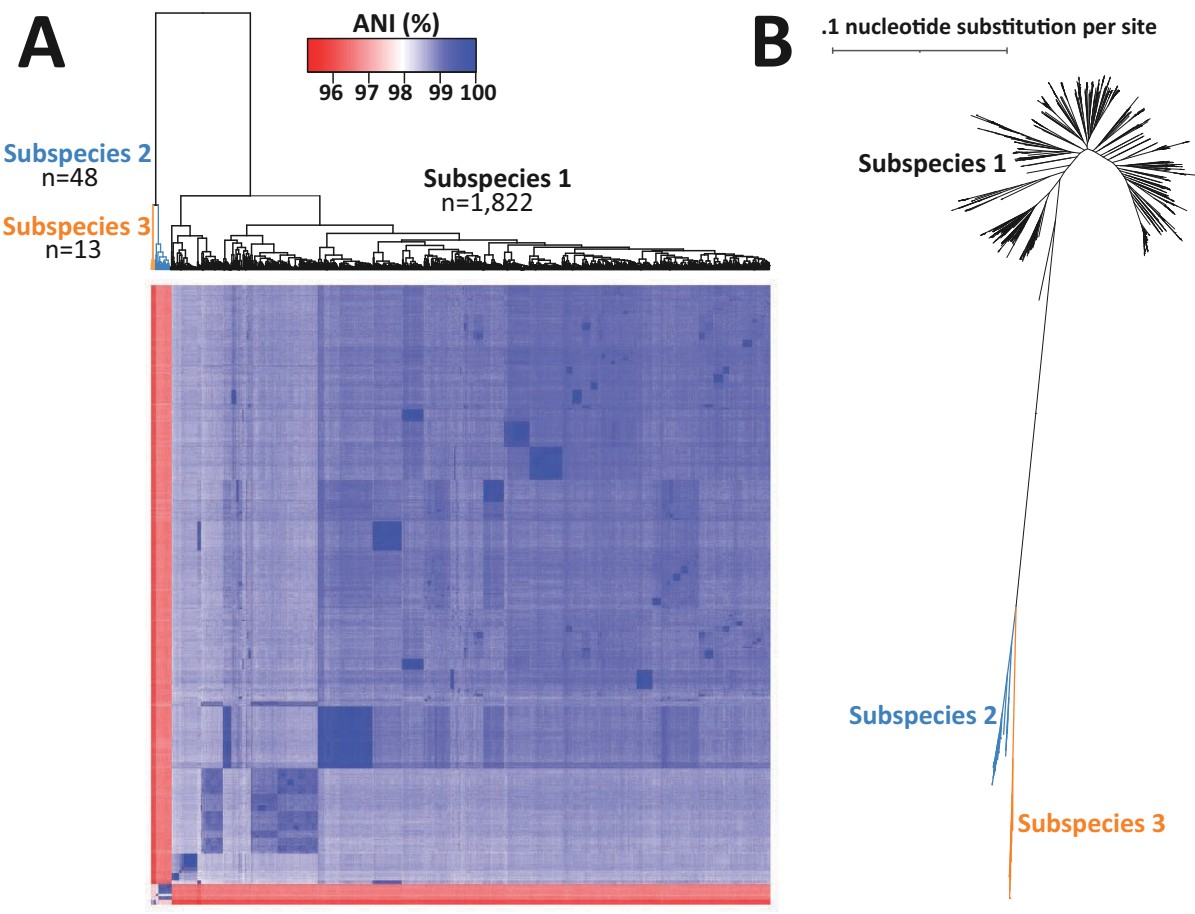

**FIG 2** *P. mirabilis* is composed of three subspecies. (A) Heat map with hierarchical clustering of 1,883 pairwise ANI values. Dendrogram is colored by subspecies assignment—reciprocal groupings of ≥98% within the group and ≤98% between groups. (B) SNP core genome phylogenetic tree with subspecies designation as colored branches.

cohort (Fig. 4A). When observed unrooted, the phylogenetic tree assumes a "star-comet" configuration, conveying that this large cohort contains both clusters of highly related genomes and substantial differences between lineages. To identify highly related clonal lineages, we quantified SNP differences between all possible pairwise genome comparisons from the SNP core genome alignment used to generate the phylogenetic tree. The mean SNP count was 14,375 with a standard deviation of 5,452 (Fig. S3A). The maximum SNP distance observed was 122,441 SNPs, and the lowest was 0. We chose a cutoff of 4,013 SNPs (equivalent to 0.1% of the median *P. mirabilis* genome length in NCBI) to specify clusters of highly related genomes. We found that the largest cluster comprised 173 genomes, while the next two largest clusters each contained 99 genomes. In total, 75 clusters of ≥ four genomes as well as 24 triplets, 48 pairs, and 196 singletons make up the cohort. The largest 16 clusters together comprise 50.9% (891/1,748) of the genomes (Fig. S3B). We compared our cluster delineation to FastBAPS group annotation as an orthogonal method to identify highly related genomes. We found that FastBAPS binned all 1,748 genomes into 31 groups, with high concordance between the two methods for the large SNP clusters (Fig. S4; Table S3). Our top 10 largest SNP clusters (*n* = 720 genomes) corresponded almost exactly with 10 BAPS groups (*n* = 730 genomes).

We layered relevant metadata for cluster assignment, anatomic source, and genome source onto the SNP core genome phylogenetic tree (Fig. 4A). In the outermost ring, we marked genomes making up the 10 largest clusters. Of note, the cluster assignment fit with the topography of our phylogenetic tree, as cluster assignments aligned with portions of the tree featuring short branch length between genomes. The top 10 largest

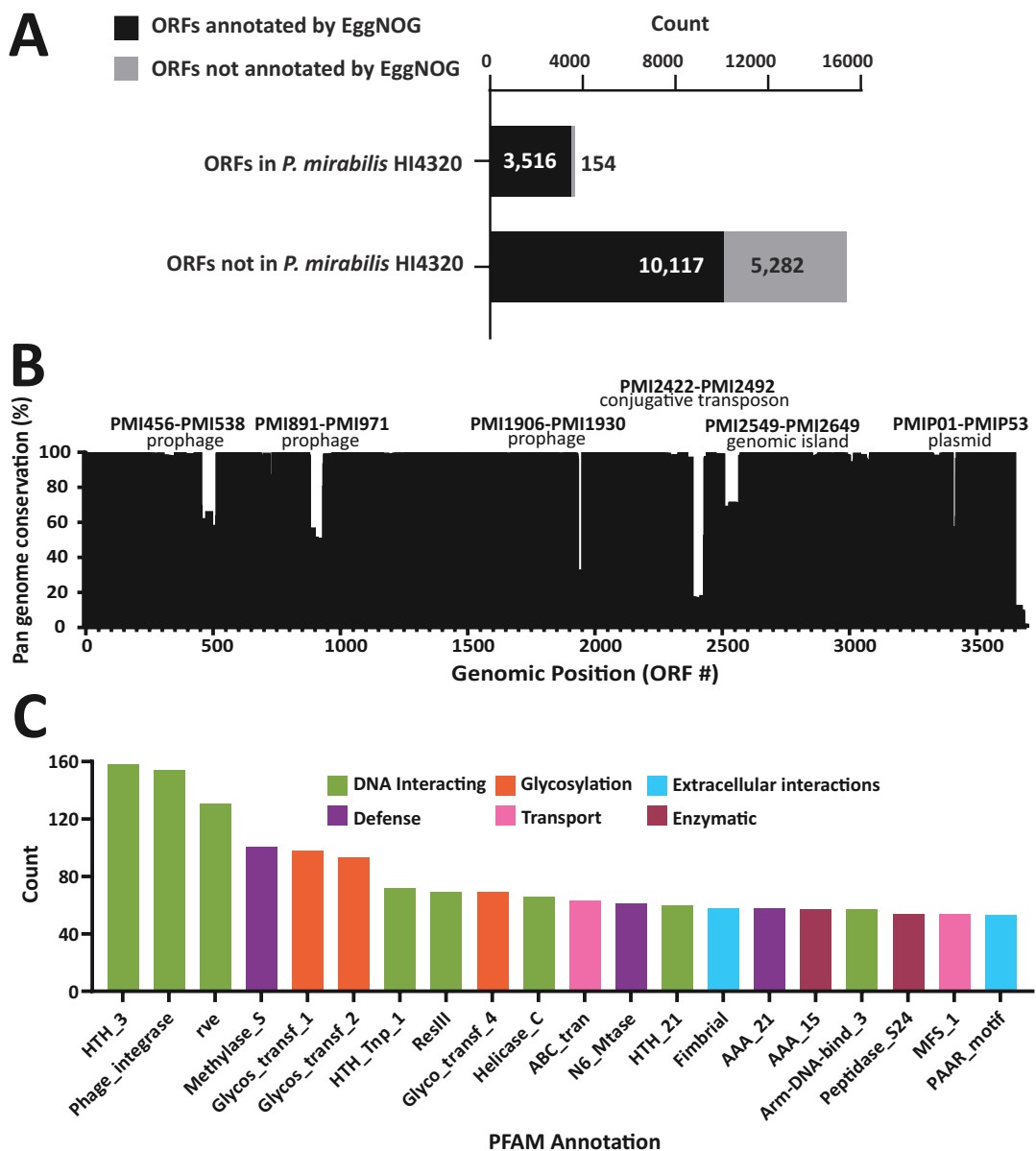

**FIG 3** *P. mirabilis* encompasses a vast accessory genome beyond *P. mirabilis* HI4320. (A) Count of genes present in *P. mirabilis* HI4320 or not present in *P. mirabilis* HI4320, analyzed by whether the genes were annotated by EggNOG-mapper. (B) Relationship between the chromosomal position of *P. mirabilis* HI4320 genes (x-axis, by prokka gene number) and prevalence of each gene within the pan-genome (y-axis). (C) Top 20 pfam annotations of the 10,117 genes annotated by EggNOG-mapper but not present in *P. mirabilis* HI4320.

clusters were spread across the phylogenetic tree with Clusters one, six, and three composing one branch (purple) and Clusters four, five, nine, and seven constituting another branch (teal, Fig. 4A). We found *P. mirabilis* HI4320 was not within the top 10 largest clusters but did have four other genomes related to it below our 4,013 SNP cutoff. We did not find any statistically significant associations between the anatomic sources and isolate cluster, indicating that clusters of highly related *P. mirabilis* genomes were capable of surviving in an array of distinct human host niches. Similarly, there were no large clades associated with the geographic (our study) or electronic (NCBI Assembly or SRA) source of the genomes. A relative exception was Cluster one, in which 84.9% (147/173) of the genomes were from NCBI Assembly but only 7.5% (13/173) were from our newly sequenced isolates. These data indicate our cohort is representative of *P.*

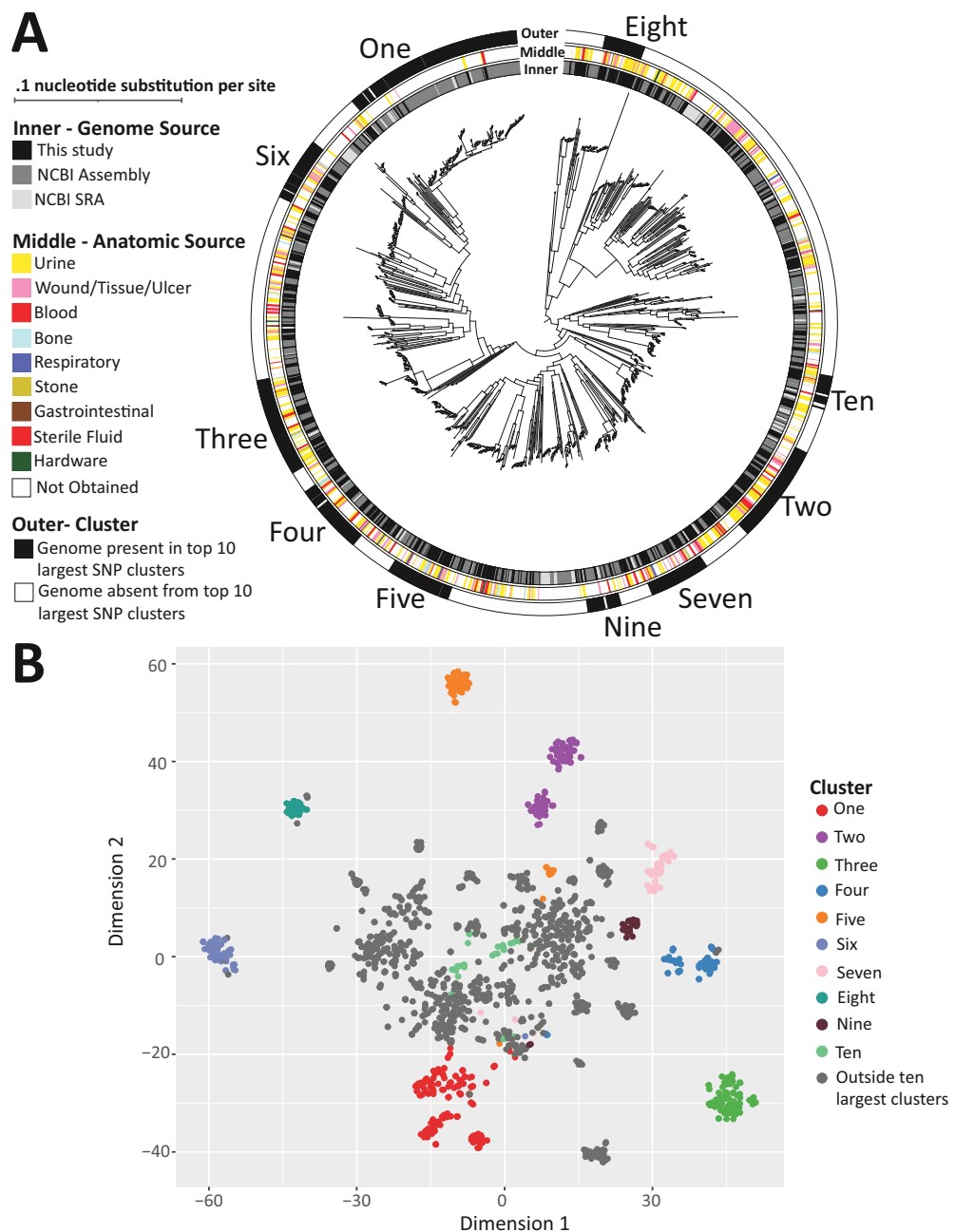

**FIG 4** *P. mirabilis* subspecies 1 comprises several large, highly related clusters with linked accessory gene content. (A) SNP core genome phylogenetic tree of 1,748 *P. mirabilis* subspecies 1 genomes, surrounded by three rings depicting genomic source, anatomic source, and presence in one of the 10 largest clusters (numbered one through ten according to cluster size). (B) t-SNE plot in which each point depicts the accessory gene content of a single genome. Points are colored by the presence in the 10 largest clusters.

*mirabilis* genomic diversity at-large. Using t-SNE analysis on a reduced presence/absence matrix representing all non-core, non-singleton genes, we found the different clusters occupied distinct spaces within the plot, indicating the correlation between phylogenetic position and accessory gene content (Fig. 4B). In addition, we overlaid the newick tree for subspecies 1 genomes (distance not to scale) adjacent to the presence/absence matrix, with genes ordered from most to least conserved (Fig. S5). We can observe that genomes within the top 10 largest SNP clusters have similar gene presence/absence

profiles to other genomes within the cluster but are distinct in comparison to the nearest neighboring genomes outside of the cluster (Fig. S5).

## Lineage-specific genes include prophages, secretion systems, and O-antigen modification

To identify genes enriched in each of the three largest clusters of subspecies 1, we performed a pan-genome-wide association study (PGWAS) using scoary. We filtered results to include only genes with ≥80% sensitivity and specificity. Due to our interest in groups of spatially co-localized genes (candidate operons), we used complete genomes to assign genomic position (Table S3; Document S2). Distinct co-localized genes annotated by Egg-NOG mapper as phage related were significantly associated (bonferroni adjusted $P < 0.001$) with Clusters one, two, and three (Fig. 5A through C). For example, distinct prophages in Cluster one encode enzymes necessary for the synthesis of queuosine (a modified nucleoside in tRNA) as well as *flu*, encoding antigen 43, a type Va secretion system important for autoaggregation in *Escherichia coli* (38) (Fig. 5A). Genes with pfam annotations for PAAR_motif and RHS_repeat, typically associated with type six secretion system proteins, were found in Cluster one and Cluster three (Fig. 5A and C). We also found a group of genes associated with Clusters two and three that are likely involved in O-antigen modification (Fig. 5B and C). The putative O-antigen modification region in Clusters two and three both contain a glycosyltransferase family 4 protein (NCBI Reference Sequence: WP_004249908.1) and a UDP glucuronic acid epimerase (NCBI Reference Sequence: WP_263055654.1) but an additional eight genes are specific to Cluster two and five genes to Cluster three.

## Uncharacterized *P. mirabilis* candidate genes acting at the host-microbe interface

We used a homology-based approach to identify gene clusters in *P. mirabilis* subspecies 1 that have been described in other Gram-negative bacteria but are not harbored in *P. mirabilis* HI4320 (Fig. 6). We quantified the distribution of a putative type IV pilus, a chaperone-usher pilus system significantly associated with Cluster three (Fig. 5C), an uncharacterized type one secretion system (T1SS), the type $V_a$ autotransporter antigen 43, and a recently reported but uncharacterized type X secretion system (TXSS) (39). The type IV pilus [found in 10 (0.6%) genomes], the T1SS [found in 1.6% 28 (1.6%) genomes], and the TXSS [found in 41 (2.3%) genomes] were not associated with the large clusters but present only in the aforementioned deep branching lineages between Clusters eight and ten. Antigen 43 was present in 63.6% (110/173) of Cluster one genomes and 76.7% (46/60) of Cluster six genomes. Unlike the other genes highlighted in this section, antigen 43 was also sporadically distributed in smaller numbers throughout the phylogenetic tree.

## DISCUSSION

To the best of our knowledge, this is the largest cohort of *Proteus* genomes analyzed. We initially found that *P. mirabilis*, *P. myxofaciens*, and *P. hauseri* represent lineages separate from the remainder of *Proteus* species and genomospecies. This informed understanding of *Proteus* phylogeny is helpful for investigating the relationship between important phenotypic features that are routinely used by clinical microbiology laboratories for identification of the members of this genus, specifically the association of indole production with an inducible chromosomal Class C beta-lactamase (40). This feature is absent in *P. penneri* and *P. mirabilis* but present in other species such as *P. vulgaris* (40). ANI analysis revealed the presence of three different *P. mirabilis* subspecies using the proposed 98% ANI cutoff from *Salmonella* subspecies (27). Further phenotypic characterization of representative isolates from the three *P. mirabilis* subspecies is necessary to create named subspecies designations. In *Salmonella*, the breakdown into subspecies categories was predictive of chaperone-usher pilus repertoire (41). *Mycobacterium*

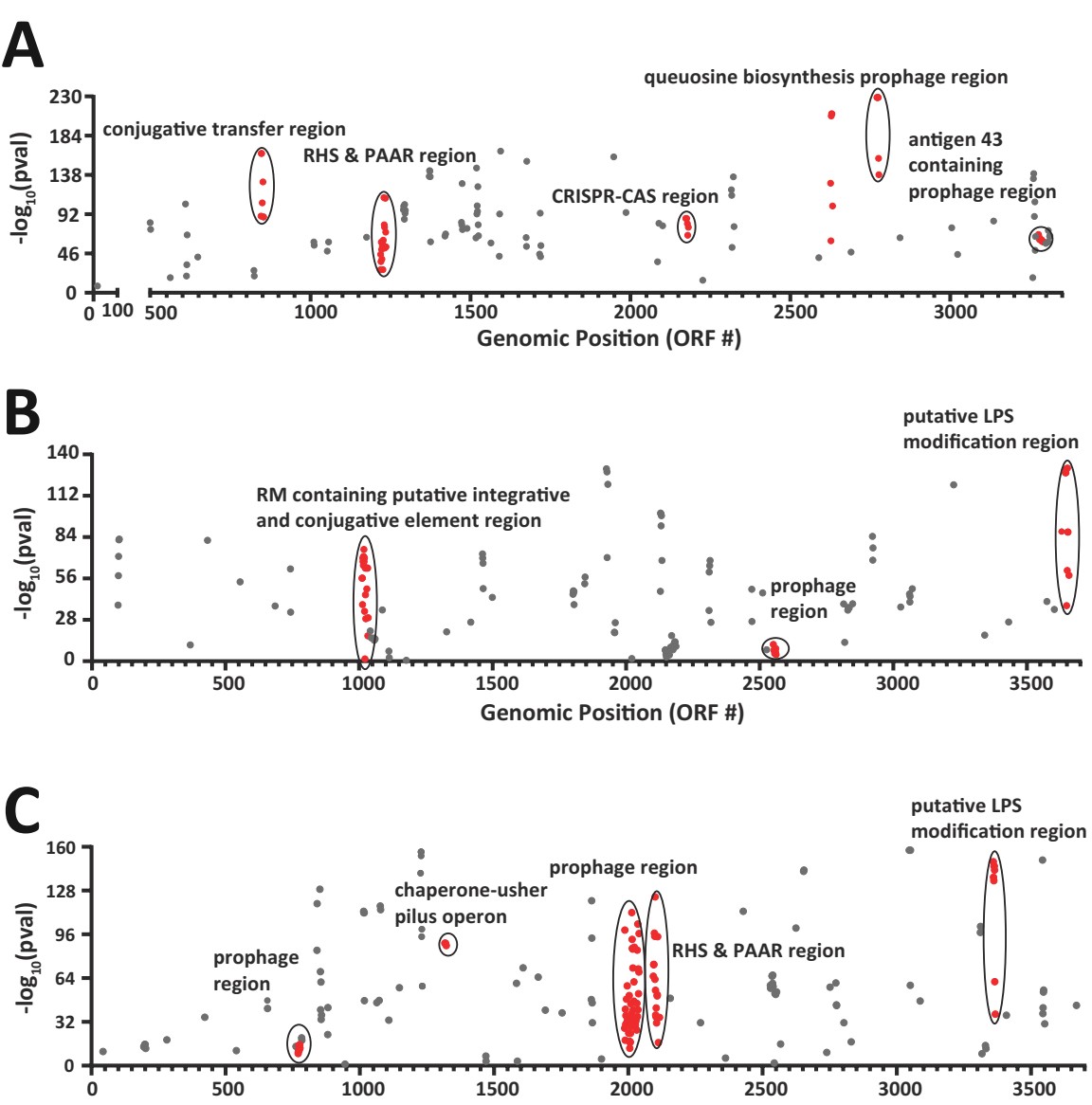

**FIG 5** PGWAS identifies gene groupings with putative localization at host-microbe interface. Manhattan plots showing genes significantly associated with the presence in Cluster one (A), Cluster two (B), and Cluster three (C). Genes depicted are only those with ≥80% both sensitivity and specificity. Genes in large consecutive (>six ORFs) groups are colored red.

*abscessus* has been historically well characterized into *M. abscessus* subspecies *bolletii*, *M. abscessus* subspecies *abscessus*, and *M. abscessus* subspecies *massiliense*. Analysis of 1,505 *M. abscessus* genomes found that 63% were *M. abscessus* subsp. *abscessus,* 30% were *M. abscessus* subsp. *massiliense,* and 7% were *M. abscessus* subsp. *bolletii* (25). In contrast, our tripartite subspecies breakdown of *P. mirabilis* was more skewed in its distribution, with nearly 97% of the genomes in subspecies 1, 2.5% in subspecies 2, and 0.7% in subspecies 3. This may reflect sampling bias in our human-associated isolates.

The use of whole-genome sequencing bacterial cohorts to investigate niche specificity and infectious capabilities is an active area of research, with often species specific observation. For our cohort, within subspecies 1 genomes, we identified highly related clusters of ≤4,013 SNPs and found that accessory genome content was related to cluster designation. This result is analogous to observations made in *E. coli* and *Enterococcus faecalis* showing the linkage between phylogenetic signal and the

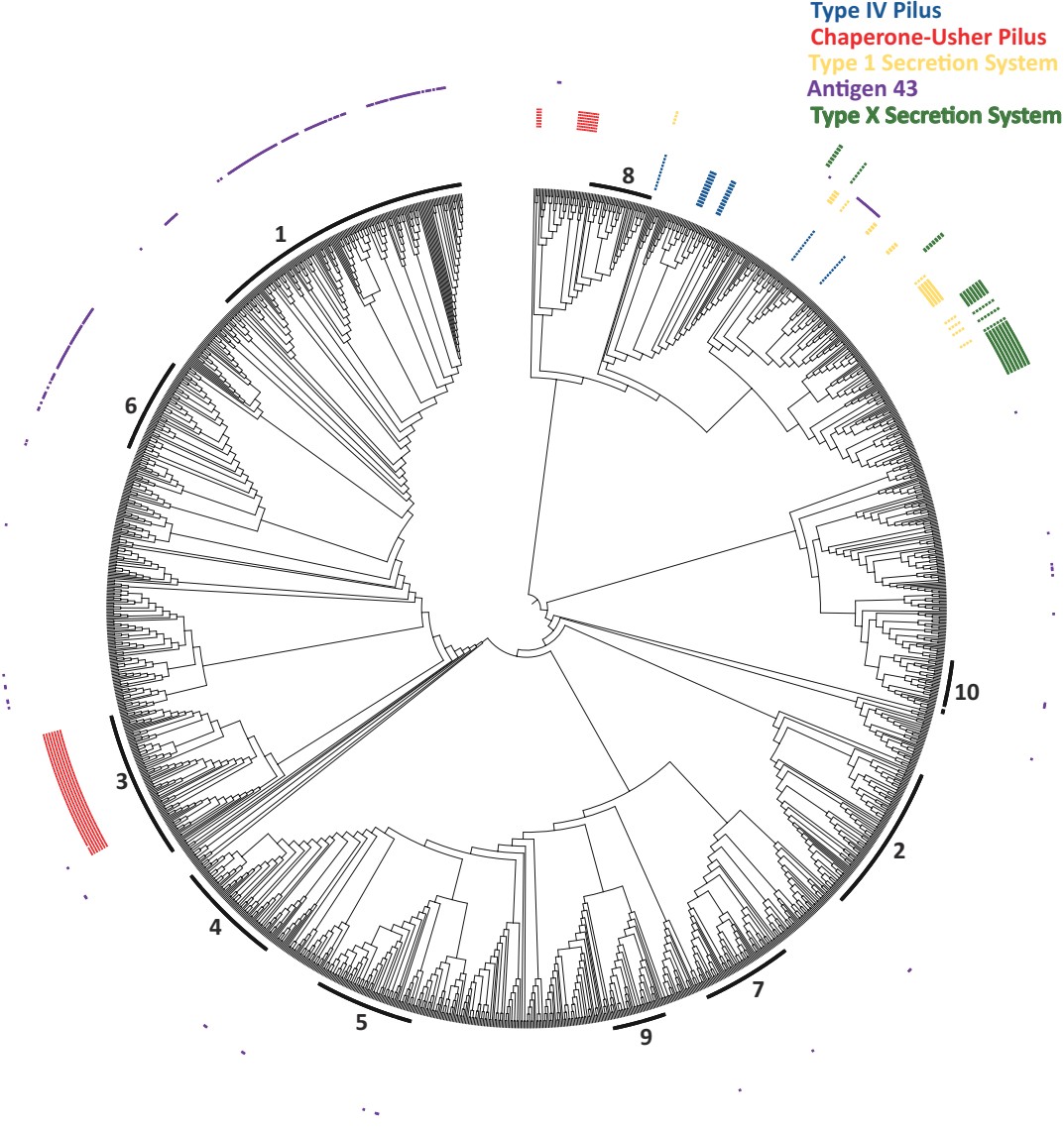

**Type IV Pilus**
**Chaperone-Usher Pilus**
**Type 1 Secretion System**
**Antigen 43**
**Type X Secretion System**

**FIG 6** Identification and distribution of uncharacterized *P. mirabilis* genes. Dendrogram of the SNP core genome phylogenetic tree with cluster position shown as an inner ring. Presence of genes is denoted by filled squares.

accessory genome (42, 43). We found that *P. mirabilis* subspecies 1 strains isolated from the same body site can come from a variety of clusters, similar to observation from a cohort of 568 avian pathogenic *E. coli* strains multiple clades can cause colonization or invasive disease (44). In addition, the authors did not identify differential plasmid burden among colonizing and disease-causing strains (44). Analysis of 490 *Camplyobacter jejuni* genomes, from human, animal, and environmental sources did find 49 accessory genes associated with pig colonization (45). *C. jejuni* is a strictly gastrointestinal pathogen in contrast to the generalist nature we found for *P. mirabilis*, suggesting that different selective pressure may be exerted on bacteria based on their ability to colonize multiple sites within an organism (45).

In our investigation, we highlighted just a few examples of such genes using PGWAS, association- and homology-based approaches. We provided the first description of antigen 43 in *P. mirabilis*, an autotransporter which in *E. coli* confers aggregation, promotes biofilm formation, and possibly prolongs cystitis (38, 46). Within *E. coli*, antigen 43 can be subdivided into four variants, but all of the *P. mirabilis* alleles in the present

study were C4 (38). While a previously characterized type Va secretion system with protease activity (*pta*) has a demonstrated role in bladder colonization, antigen 43 is a self-associated autotransporter with mechanistically distinct biological activity (47). PGWAS also identified a chaperone-usher pilus associated with Cluster three; such systems are abundant in other Gram-negative bacteria and in *E. coli* mediate binding to various glycoprotein targets on uroepithelial cells (48). In *K. variicola* and *Salmonella*, chaperone-usher pili were similarly associated with phylogenetic position rather than environmental source (41, 49). *P. mirabilis* HI4320 has 17 chaperone-usher pilus systems, with 5/17 having a demonstrated biological role (50). The existence of cluster associated accessory chaperone-usher pilus could confer increased adherence of *P. mirabilis* isolates in specific environmental niches. In *K. pneumoniae*, the Kpi chaperone-usher pilus was associated with presence in the ST-15 conferred increased adherence to alveolar, bladder, and colorectal cell lines (8). We also found that genes putatively associated with O-antigen synthesis were significantly enriched within Cluster two and Cluster three genomes. This is consistent with previous genetic characterization of 80 *Proteus* genomes using PCR which found 60 distinct O-antigen serotypes. In *Pseudomonas aeruginosa*, a Gram-negative pathogen also capable of causing disease in a variety of infectious contexts, genetic modifications within isogenic strains that caused modification to O-antigen were associated with decreased inflammation during chronic infection (51). Further work can precisely understand the relationship between complete LPS structure and O-antigen gene presence as well as the impact the different LPS structures may have on host interactions. Using a homology-based approach we scoured the pan-genome for other genes whose products may act at the host-microbe interface. We found the recently described TXSS, which encodes a tripartite toxin uncharacterized in *P. mirabilis*, was rare and not cluster associated (39). Similar to our description in *P. mirabilis*, *Yersinia enterocolitca* strain W22703 encodes a TXSS and deletion of different components attenuates virulence in a systemic *Galleria mellonella* (Greater wax moth) infection model (52). The operon surrounding the cytotoxin gene *tcsL* in *Paeniclostridium sordellii* also encodes components similar to the TXSS, suggesting that this might be a conserved export mechanism for toxins. Further work using reverse genetics can characterize the role of this secretion system in *P. mirabilis*. The T1SS we identified carries an effector with multiple bacterial immunoglobulin domains, which are typically found in adhesins (53). Further work using reverse genetic approaches previously developed for *P. mirabilis* will expand on our genomic findings to better understand how specific accessory genes may impact bacterial physiology and pathogenicity.

In conclusion, we performed whole-genome based analysis on the largest cohort of *Proteus* spp. to date. A limitation of our study is the lack of isolates from healthy people (which may represent true commensalism) or from animal or environmental samples. Additionally, since we used Illumina whole-genome sequencing methodology, we were not able to confidently separate complete chromosomes from plasmids. We highlighted inter-species diversity of the genus as well as intraspecies diversity within *P. mirabilis*. We found that subspecies 1 can be further divided into highly related clusters with linkage between clusters and accessory genome composition. Notable accessory genes include lineage associated prophage regions, chaperone-usher pili, a type V secretion system as well as a lineage independent novel T1SS, and an uncharacterized TXSS. This repertoire of accessory genome components may lead to strain specific mechanisms of interactions at the host-microbe interface during colonization and pathogenesis. Phenogenomics—the combined use of profiling phenotypic variation with genomic annotation has identified a novel secretion mechanism in *M. abscessus* and *Verrucosispora* sp. adaption to laboratory growth conditions (54, 55). The whole-genome sequenced clinical isolate library from this study is a necessary initial step using phenogenomics to reveal how SNP variation in core genes or accessory gene presence/absence influences strain behavior across a variety of model systems. These results argue strongly that the nearly exclusive use of a single model strain impedes our ability to understand how other genetic factors may influence infection dynamics.

## ACKNOWLEDGMENTS

The authors thank members of the Rosen, Yarbrough, Burnham, Dantas, and Hunstad laboratories for insightful discussion. The authors thank the Barnes-Jewish Hospital clinical microbiology laboratory technologists for their enthusiastic Proteus plate saving.

R.F.P. received project support from the Academy of Clinical Laboratory Physicians and Scientists Young Investigator Grant Program and salary support from American Urologic Association Research Scholar Program. This work was supported in part by the National Institute of Allergy and Infectious Diseases of the NIH (grant numbers U01AI123394 and R01AI155893 to G.D. and R01AI158418 to D.A.H.) and the Agency for Healthcare Research and Quality (grant number R01HS027621). The content is solely the responsibility of the authors and does not necessarily represent the official views of the funding agencies.

## AUTHOR AFFILIATIONS

[1]Department of Pediatrics, Washington University School of Medicine in St. Louis, St. Louis, Missouri, USA

[2]Department of Pathology & Immunology, Washington University School of Medicine in St. Louis, St. Louis, Missouri, USA

[3]The Edison Family Center for Genome Sciences & Systems Biology, Washington University School of Medicine in St. Louis, St. Louis, Missouri, USA

[4]Department of Pathology and Laboratory Medicine, Weill Cornell Medicine, New York, New York, USA

[5]Department of Pathology and Laboratory Medicine, NorthShore University Health System, Evanston, Illinois, USA

[6]Department of Molecular Microbiology, Washington University School of Medicine in St. Louis, St. Louis, Missouri, USA

[7]Department of Biomedical Engineering, Washington University in St. Louis, St. Louis, Missouri, USA

[8]Department of Medicine, Washington University School of Medicine in St. Louis, St. Louis, Missouri, USA

## AUTHOR ORCIDs

Robert F. Potter  http://orcid.org/0000-0002-4987-7634
Carol E. Muenks  http://orcid.org/0000-0003-4294-8134
Melanie L. Yarbrough  http://orcid.org/0000-0003-2166-7552
David A. Hunstad  http://orcid.org/0000-0002-9848-0975
Gautam Dantas  http://orcid.org/0000-0003-0455-8370
Carey-Ann D. Burnham  http://orcid.org/0000-0002-1137-840X

## FUNDING

| Funder | Grant(s) | Author(s) |
| --- | --- | --- |
| Urology Care Foundation (UCF) | Research Scholar | Robert F. Potter |

## AUTHOR CONTRIBUTIONS

Kailun Zhang, Data curation, Writing – review and editing | Ben Reimler, Data curation | Jamie Marino, Data curation | Carol E. Muenks, Data curation, Writing – review and editing | Kelly Alvarado, Data curation | Meghan A. Wallace, Data curation | Lars F. Westblade, Data curation | Erin McElvania, Data curation | David A. Hunstad, Writing – review and editing, Funding acquisition, Conceptualization | Gautam Dantas, Conceptualization, Data curation, Funding acquisition, Writing – review and editing | Carey-Ann D. Burnham, Conceptualization, Investigation, Supervision, Writing – review and editing.

## DATA AVAILABILITY STATEMENT

Adapter-removed Illumina reads and scaffolds for genomes generated in this study have been deposited to NCBI SRA and NCBI Assembly, respectively. The associated NCBI BioProject ID is PRJNA893581.

## ETHICS APPROVAL

Access to the patient laboratory information system (Cerner Millennium) and EMR (EPIC) for the Washington University School of Medicine was approved by the Institutional Review Board (IRB Approval #202011036). Identifying health information was not obtained for samples from NorthShore University or Weill Cornell Medicine, so IRB approval was not sought.

## ADDITIONAL FILES

The following material is available online.

### Supplemental Material

**Figure S1 (mSystems00159 S0001.eps).** ANI Clusters.
**Figure S2 (mSystems00159 S0002.eps).** Pan genome statistics.
**Figure S3 (mSystems00159 S0003.eps).** Cluster rarefaction.
**Figure S4 (mSystems00159 S0004.eps).** SNP Cluster and FastBAPS overlay.
**Figure S5 (mSystems00159 S0005.tif).** Core genome phylogenetic tree and gene presence/absence matrix.
**Document S1 (mSystems00159 S0006.docx).** Commands used in this study.
**Document S2 (mSystems00159 S0007.txt).** Fasta formatted file for specific genes identified in pan-genome.
**Tables S1 to S5 (mSystems00159 S0008.xlsx).** Genomes used, QUAST quality statistics, genome designation within FastBAPS groups and SNP Clusters, co-localized genes identified by scoary as cluster associated, and genes portrayed in Figure 6.

### Open Peer Review

**PEER REVIEW HISTORY (review-history.pdf).** An accounting of the reviewer comments and feedback.

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
