## [Reviewer comments · mSystems]

Uncharacterized and lineage specific accessory genes within the *Proteus mirabilis* pan-genome landscape

Robert Potter, Kailun Zhang, Ben Reimler, Jamie Marino, Carol Muenks, Kelly Alvarado, Meghan Wallace, L Westblade, Erin McElvania, Melanie Yarbrough, David Hunstad, Gautam Dantas, and Carey-Ann Burnham

Corresponding Author(s): Robert Potter, Washington University in St Louis School of Medicine

Review Timeline:

Submission Date:	February 15, 2023
Editorial Decision:	March 20, 2023
Revision Received:	May 3, 2023
Accepted:	May 7, 2023

Editor: Sean Gibbons

Reviewer(s): The reviewers have opted to remain anonymous.

Transaction Report:

DOI: <https://doi.org/10.1128/msystems.00159-23>

March 20, 2023

Dr. Robert F. Potter
Washington University in St Louis School of Medicine
660 South Euclid Ave
Campus Box 8118
Saint Louis, Missouri 63110

Re: mSystems00159-23 (Uncharacterized and lineage specific accessory genes within the *Proteus mirabilis* pan-genome landscape)

Dear Dr. Robert F. Potter:

Thank you for submitting your manuscript to mSystems. Overall, the reviewers thought your work was of interest, but had a few concerns. Therefore, minor modifications are necessary before we can accept this manuscript.

Please thoroughly address all reviewer comments and concerns. In particular, please provide a better justification for your subspecies clustering, remove some of the redundancy across the results and discussion sections, and provide additional discussion of your work in the context of the broader field.

Preparing Revision Guidelines

Sincerely,

Sean Gibbons

Editor, mSystems

Journals Department

Reviewer comments:

Reviewer #1 (Comments for the Author):

Potter et al. presents their findings on the pan-genome characteristics of the opportunistic pathogen *P. mirabilis*. Using newly sequenced and previously sequenced genomes, the study described three subspecies and the variation in accessory genome content within the species. Below are my comments.

Major comments:

1. Results, lines 271-272: The rationale for choosing 4,013 SNPs = 0.1% of the median *P. mirabilis* genome length in NCBI in delineating clusters should be clarified here. There are several robust and widely used methods to infer bacterial population structure and clustering, e.g., BAPS, Mandrake, PopPUNK, so the 0.1% median definition used in this study is unclear.
2. The Discussion is weak, with more than half of the Discussion merely reiterating what was already mentioned in the Results section. The broader implications of the findings should be described.

Minor comments:

1. Intro, Lines 90-92: Reference 14 is cited here, but this reference only mentions urine cultures. A proper reference for *P. mirabilis* in wound and soft tissue infections should be cited here.
2. Methods, lines 141-142: The p value used as threshold for the chi-sq analysis should be stated here.
3. Methods, Line 172: Citation for RStudio is missing.
4. Methods, Line 174: I suggest removing "3 subspecies" in this sentence as they have not been distinguished yet at this point of the text. Alternatively, you can describe here how the three subspecies were delineated.
5. Throughout the text: spell out numbers below ten. E.g., three instead of 3 (line 174), five instead of 5 (line 248), three instead of 3 (line 295)
6. Methods, lines 155, 163: What quality metrics were used to assess the genomes? Supplementary table S1 should include information about the number of contigs, N50, number of annotated genes, genome size for each genome included in the study. Table S1 is missing the headings for each column.
7. Methods, line 169: Citation for Cytoscape is missing.
8. Methods, lines 179-180 "To gain specific insight into the population structure of subspecies 1, we used RAxML to identify identical isolates": Phylogenetic tree reconstruction using RAxML needs to include description of parameters used (e.g., bootstrapping, rooting, nucleotide substitution model, rate heterogeneity). Moreover, one does not use a phylogenetic tree to determine the population structure and identify identical isolates; hence it is unclear what this sentence means. It is unclear what the rationale is for removing the duplicate genomes.
9. Methods, line 195: Citation for Scoary is missing.
10. Results, line 226: What does "approximate" tree mean here?
11. Results and Methods: It is unclear why EGGnog was used for annotation for subspecies 1 and Prokka for the other *Proteus* genomes.
12. The figure legends associated with each figure is very confusing. For figures, legends should be placed below the figure. For example, figure 2 legend is found below and within the same page as figure 1.
13. Figure 4: Colored branches in the tree should be defined. What "within and outside" 10 largest clusters mean is not clear.
14. Results, lines 304-305: What the genes are that were mentioned as to be likely involved in O-antigen modification should be mentioned/listed here. Better yet, a supplementary table listing the gene names and functions of those identified in red dots or circled in Figure 5 should be included.

Reviewer #2 (Comments for the Author):

The authors of this study obtained a substantial number of *Proteus* isolates, which they sequenced and analyzed to gain significant insights into the population genetics of this taxon. The resulting collection of isolates, including both experimentally ready-to-use isolates and genomes, will prove invaluable in comprehending the functional implications of intra-species diversity. Such knowledge is crucial in elucidating the molecular mechanisms underlying host-bacteria interactions. The clarity of the analysis and writing in this paper is pretty good.

1. The article utilized over 1000 genomes from a publicly accessible database. The authors ought to clarify the supplementary worth of the data generated in this study in relation to existing datasets. Specifically, does the new data feature a greater

number of human donors or additional timepoints?

2. It would also be helpful to summarize how many donors are included in both the new data and the existing public data.
3. Figure 1 would benefit from improvement to enable readers to discern its details more clearly.
4. In my opinion, Figure 2A appears to display either 2 or 4 subspecies, depending on where to set the cutoff. While I acknowledge that non-supervised clustering can be subjective, the authors should furnish additional justifications for their decision to conclude that there are 3 subspecies.
5. It might be worth considering creating a new plot that utilizes SNP distance and gene presence-absence distance as the two axes in Figure 4. This would enable the authors to reinforce the conclusions derived from Figure 4 more directly.
6. In the abstract, the authors use a single paper demonstrating that intra-species diversity was once believed to be minor to motivate readers. However, it is unclear whether this notion is widely acknowledged as the prevailing understanding within the field.
7. The collection of isolates in one lab (?) is a tremendous resource for future experimental research. The authors may wish to expand upon this point in their discussion, particularly in comparison to public datasets.

Reviewer #1 (Comments for the Author):

Potter et al. presents their findings on the pan-genome characteristics of the opportunistic pathogen *P. mirabilis*. Using newly sequenced and previously sequenced genomes, the study described three subspecies and the variation in accessory genome content within the species. Below are my comments.

We thank Reviewer #1 for devoting their time and scientific insight towards our manuscript and agree with their summary statement.

Major comments:

1. Results, lines 271-272: The rationale for choosing 4,013 SNPs = 0.1% of the median *P. mirabilis* genome length in NCBI in delineating clusters should be clarified here. There are several robust and widely used methods to infer bacterial population structure and clustering, e.g., BAPS, Mandrake, PopPUNK, so the 0.1% median definition used in this study is unclear.

We agree with the reviewer that this important detail within our manuscript should have additional explanation. We did not initially use a bacterial population structure program because we explicitly did not want to cluster all genomes. We wanted to allow for the existence of singleton and pair genomes, which is not a feature of population structure software, as this was the first foray into *P. mirabilis* phylogeny. We took the reviewer's suggestion and applied FastBAPS to our core-genome alignment of de-duplicated subspecies 1 genomes. We chose FastBAPS since it was created by the author of panaroo it is benchmarked for utility on large (>1,000) genome datasets. FastBAPS identified 31 clusters, much smaller than our count of 75 clusters (containing ≥ 4 genomes), however we had great concordance between what we identified as the top 10 largest clusters. We identified 720 genomes within the top 10 largest clusters, while FastBAPS identified 730 genomes in the corresponding BAPS groups. All 720 of the top 10 largest cluster genomes are within the 730 identified by FastBAPS. We investigated the 10 discrepant genomes. Four genomes were identified in BAPS group 20 but not in SNP cluster 1, three genomes were identified in BAPS group 23 but not SNP cluster 4, one genome was identified in BAPS group 14 but not SNP cluster 5, one genome was identified in BAPS group 3 but not SNP cluster 8, and one genome was identified in BAPS group 17 but not SNP cluster 9. The discrepant genomes are the white rectangles within the respective SNP clusters, indicating they are closely related to other genomes in the cluster but with $\geq 4,013$ SNP distance. Given the high concordance between FastBAPS groupings and SNP clusters, we have confidence in our original downstream interpretation of accessory genome content distribution. We have included a visualization of this concordance as Figure S4 and all of the results of FastBAPS groups and SNP clusters as Table S3. Lines 186-192, 286-290.

2. The Discussion is weak, with more than half of the Discussion merely reiterating what was already mentioned in the Results section. The broader implications of the findings should be described.

We appreciate the reviewer's insight into our Discussion section and have addressed this by removing most of the reiterations of our own results, and by expanding on the broader context of our work within the field of Gram-negative pan-genomics and on specific genes of interest identified in the pan-genome but absent from *P. mirabilis* HI4320. Lines 366-373, 384-396, 398-404, 418-424.

Minor comments:

1. Intro, Lines 90-92: Reference 14 is cited here, but this reference only mentions urine cultures. A proper reference for *P. mirabilis* in wound and soft tissue infections should be cited here.

We have added a reference for a study identifying the microbiology of urine and wound cultures and found that *P. mirabilis* is a major component of both (PMID: 31462413). Lines 81-84.

2. Methods, lines 141-142: The p value used as threshold for the chi-sq analysis should be stated here.
We have clarified that a threshold p of .05 was used for significance testing. Lines 133-135.

3. Methods, Line 172: Citation for RStudio is missing.

We thank the reviewer for noticing that we did not properly credit RStudio and have added a citation to remedy this. Lines 166.

4. Methods, Line 174: I suggest removing "3 subspecies" in this sentence as they have not been distinguished yet at this point of the text. Alternatively, you can describe here how the three subspecies were delineated.

We thank the author for their helpful comment and took both approaches. We have removed "3 subspecies" and kept the meaning of the sentence the same as well as adding more information on methods in the concluding portion of the previous paragraph. Lines 166-169.

5. Throughout the text: spell out numbers below ten. E.g., three instead of 3 (line 174), five instead of 5 (line 248), three instead of 3 (line 295)

We have changed numerical values below 10 into their word forms. Throughout text.

6. Methods, lines 155, 163: What quality metrics were used to assess the genomes? Supplementary table S1 should include information about the number of contigs, N50, number of annotated genes, genome size for each genome included in the study. Table S1 is missing the headings for each column.
We have included a description of our quality cutoff being <500 contigs as well as including Table S2 as a supplemental file. Line 149.

7. Methods, line 169: Citation for Cytoscape is missing.

We have added the appropriate reference for Cytoscape (14597658). Line 163.

8. Methods, lines 179-180 "To gain specific insight into the population structure of subspecies 1, we used RAxML to identify identical isolates": Phylogenetic tree reconstruction using RAxML needs to include description of parameters used (e.g., bootstrapping, rooting, nucleotide substitution model, rate heterogeneity). Moreover, one does not use a phylogenetic tree to determine the population structure and identify identical isolates; hence it is unclear what this sentence means. It is unclear what the rationale is for removing the duplicate genomes.

We thank the author for pointing out this confusion. We have clarified that we used RAxML only to identify duplicate genomes from the initial core-genome alignment of subspecies 1 genomes (n=1,883). We have added our rationale for this in the text and below. We have removed the phrase "To gain specific insight into the population structure of subspecies..." Lines 176-180.

9. Methods, line 195: Citation for Scoary is missing.

We have added the reference for Scoary in the methods section. Line 203.

10. Results, line 226: What does "approximate" tree mean here?

The use of “approximate” comes from FastTree 2 documentation which frequently adds the qualifier in front of language describing the algorithm in comparison to traditional maximum likelihood methods (<http://www.microbesonline.org/fasttree/>).

11. Results and Methods: It is unclear why EGGnog was used for annotation for subspecies 1 and Prokka for the other *Proteus* genomes.

We thank the reviewer for pointing out this ambiguity. We clarified prokka was used for gene calling and initial annotation in all of the *P. mirabilis* genomes in the methods but that EggNOG was only used for subspecies 1. In our experience we cannot completely trust prokka annotation, so we use EggNOG to achieve more comprehensive identification on genes called by prokka. Lines 149, 194-195.

12. The figure legends associated with each figure is very confusing. For figures, legends should be placed below the figure. For example, figure 2 legend is found below and within the same page as figure 1.

We agree with the reviewer and hope that following mSystems directions for uploading the figures as separate files has alleviated that annoyance.

13. Figure 4: Colored branches in the tree should be defined. What "within and outside" 10 largest clusters mean is not clear.

We thank the reviewer for noticing these detracting aspects of Figure 4A. We have changed all branch colors to be black and have edited the referenced boxes to say “Genome present in top 10 largest SNP clusters” and “Genome absent from top 10 largest SNP clusters” which will add clarity for the reader. Figure 4A.

14. Results, lines 304-305: What the genes are that were mentioned as to be likely involved in O-antigen modification should be mentioned/listed here. Better yet, a supplementary table listing the gene names and functions of those identified in red dots or circled in Figure 5 should be included.

We agree with the reviewer that this would be a nice piece of information useful for genomic researchers interested in *Proteus* or accessory genome functional capacities. We have added a supplementary table that lists all of the gene names (from panaroo) and their annotation (from EggNOG), as well as scoary metrics for association and an additional text file containing the representative ORF clustered by panaroo for each gene call (Table S4, Document S2). Additionally, we have mentioned two of the genes in the results section and expanded the Discussion to include thoughts on relationship to previous studies on O-antigen variability. Lines 325-329, 388-396.

Reviewer #2 (Comments for the Author):

The authors of this study obtained a substantial number of *Proteus* isolates, which they sequenced and analyzed to gain significant insights into the population genetics of this taxon. The resulting collection of isolates, including both experimentally ready-to-use isolates and genomes, will prove invaluable in comprehending the functional implications of intra-species diversity. Such knowledge is crucial in elucidating the molecular mechanisms underlying host-bacteria interactions. The clarity of the analysis and writing in this paper is pretty good.

We thank the reviewer for their feedback on our manuscript and are pleased with their interpretation for this genomic analysis as well as the broader scope of what we hope to accomplish with this cohort.

1. The article utilized over 1000 genomes from a publicly accessible database. The authors ought to clarify the supplementary worth of the data generated in this study in relation to existing datasets. Specifically, does the new data feature a greater number of human donors or additional timepoints?

The data (genomes) from the publicly accessible databases (NCBI Assembly and NCBI SRA) had been deposited but not collectively analyzed before. The use of publicly accessible genomes was helpful for contextualizing our results within the broadest possible *P. mirabilis* phylogenomic space. For instance, two outcomes of using publicly available genomes are that we now know we possess 12/13 of subspecies three isolates in existence, as only one existed in the publicly accessible databased prior to this investigation (Figure 2A) and that while cluster 1 is the largest cluster within *P. mirabilis* subspecies 1, it contains very few of our isolates relative to clusters 2 and 3 (Figure 4A). All sources from our study are human. An important limitation of our study is that we do not include non-human associated *Proteus* isolates, as mentioned in Lines 409-410.

2. It would also be helpful to summarize how many donors are included in both the new data and the existing public data.

We agree with the reviewer that knowing how many donors would be helpful for contextualizing microbial genomics results. In our study we have included genomes that come from ~702 donors. However, to the best of our knowledge there is not an intuitive way to pull out such metadata from NCBI.

3. Figure 1 would benefit from improvement to enable readers to discern its details more clearly.

We agree with the reviewer that Figure 1 details can be hard to parse out. We have tried to remedy this using different color combinations for the background, connecting lines, and nodes; however, no combination looks as intuitive as the black lines, gray shapes, on white background. We have not been able to increase the resolution of the figure in Cytoscape as this causes the app to freeze. However, given that the most important part of Figure 1 is that it shows the different *Proteus* ANI genomospecies, we have swapped it with Figure S1, which illustrates the same point in a way that is easier to parse.

4. In my opinion, Figure 2A appears to display either 2 or 4 subspecies, depending on where to set the cutoff. While I acknowledge that non-supervised clustering can be subjective, the authors should furnish additional justifications for their decision to conclude that there are 3 subspecies.

Given that this concern was shared also by Reviewer 1, we hoped to address both points by including more information on the methods for how subspecies delineation was created and including more references for the application of this cutoff in other bacterial species. Lines 165-169.

5. It might be worth considering creating a new plot that utilizes SNP distance and gene presence-absence distance as the two axes in Figure 4. This would enable the authors to reinforce the conclusions derived from Figure 4 more directly.

We agree with the reviewer that this type of visualization would reinforce our identification that *P. mirabilis* subspecies 1 clusters have similar gene content within a cluster but distinct gene content between clusters. We have modified python scripts from roary plots (https://github.com/sanger-pathogens/Roary/blob/master/contrib/roary_plots/roary_plots.ipynb) to do this. We have included this visualization as Figure S5 and believe that it portrays this observation well (Lines 307-312, 205-208).

6. In the abstract, the authors use a single paper demonstrating that intra-species diversity was once believed to be minor to motivate readers. However, it is unclear whether this notion is widely acknowledged as the prevailing understanding within the field.

We thank the reviewer for noting this point on the novelty of our study. We agree that it is not clear if this notion is widely acknowledged, as there have not been any attempts to adequately survey diversity of *P. mirabilis*, at least on a multi-center level, until our present manuscript. However, since the manuscript cited is a recent (2018) seminal review from the leading scientists in the *P. mirabilis* field, we do believe it captures a prevailing view within the field.

7. The collection of isolates in one lab (?) is a tremendous resource for future experimental research. The authors may wish to expand upon this point in their discussion, particularly in comparison to public datasets.

We agree with the reviewer that this is an important point and mention how this dataset can be used in the future as an expansion to the burgeoning field of reconciling genomic variation with phenotypic diversity in bacteria. Lines 420-424.

May 7, 2023

Dr. Robert F. Potter
Washington University in St Louis School of Medicine
660 South Euclid Ave
Campus Box 8118
Saint Louis, Missouri 63110

Re: mSystems00159-23R1 (Uncharacterized and lineage specific accessory genes within the *Proteus mirabilis* pan-genome landscape)

Dear Dr. Robert F. Potter:

Your manuscript has been accepted, and I am forwarding it to the ASM Journals Department for publication. For your reference, ASM Journals' address is given below. Before it can be scheduled for publication, your manuscript will be checked by the mSystems production staff to make sure that all elements meet the technical requirements for publication. They will contact you if anything needs to be revised before copyediting and production can begin. Otherwise, you will be notified when your proofs are ready to be viewed.

If you would like to submit a potential Featured Image, please email a file and a short legend to msystems@asmusa.org. Please note that we can only consider images that (i) the authors created or own and (ii) have not been previously published. By submitting, you agree that the image can be used under the same terms as the published article. File requirements: square dimensions (4" x 4"), 300 dpi resolution, RGB colorspace, TIF file format.

We recognize that the video files can become quite large, and so to avoid quality loss ASM suggests sending the video file via <https://www.wetransfer.com/>. When you have a final version of the video and the still ready to share, please send it to mSystems staff at msystems@asmusa.org.

Sincerely,

Sean Gibbons
Editor, mSystems

Journals Department
E-mail: mSystems@asmusa.org